# Adaptation of *Helicoverpa armigera* to Soybean Peptidase Inhibitors Is Associated with the Transgenerational Upregulation of Serine Peptidases

**DOI:** 10.3390/ijms232214301

**Published:** 2022-11-18

**Authors:** Pedro A. Velasquez-Vasconez, Benjamin J. Hunt, Renata O. Dias, Thaís P. Souza, Chris Bass, Marcio C. Silva-Filho

**Affiliations:** 1Departamento de Genética, Escola Superior de Agricultura Luiz de Queiroz, Universidade de São Paulo, Av. Pádua Dias, 11, Piracicaba 13418-900, SP, Brazil; 2College of Life and Environmental Sciences, Biosciences, University of Exeter, Penryn Campus, Penryn, Cornwall TR10 9FE, UK; 3Instituto de Ciências Biológicas, Universidade Federal de Goiás, Goiânia 74690-900, GO, Brazil

**Keywords:** phenotypic plasticity, germinal reprogramming, proteases, lepidoptera, epigenetics, evolution, generalist herbivore

## Abstract

Molecular phenotypes induced by environmental stimuli can be transmitted to offspring through epigenetic inheritance. Using transcriptome profiling, we show that the adaptation of *Helicoverpa armigera* larvae to soybean peptidase inhibitors (SPIs) is associated with large-scale gene expression changes including the upregulation of genes encoding serine peptidases in the digestive system. Furthermore, approximately 60% of the gene expression changes induced by SPIs persisted in the next generation of larvae fed on SPI-free diets including genes encoding regulatory, oxidoreductase, and protease functions. To investigate the role of epigenetic mechanisms in regulating SPI adaptation, the methylome of the digestive system of first-generation larvae (fed on a diet with and without SPIs) and of the progeny of larvae exposed to SPIs were characterized. A comparative analysis between RNA-seq and Methyl-seq data did not show a direct relationship between differentially methylated and differentially expressed genes, while trypsin and chymotrypsin genes were unmethylated in all treatments. Rather, DNA methylation potential epialleles were associated with transcriptional and translational controls; these may play a regulatory role in the adaptation of *H. armigera* to SPIs. Altogether, our findings provided insight into the mechanisms of insect adaptation to plant antiherbivore defense proteins and illustrated how large-scale transcriptional reprograming of insect genes can be transmitted across generations.

## 1. Introduction

*H. armigera* (Hübner) (Lepidoptera: Noctuidae) is one of the most economically important crop pests globally. This species is currently distributed across five continents [1] and is highly polyphagous, threatening more than 172 plant species [2]. The exceptional ability of *H. armigera* to utilize different habitats is due, among other factors, to its capacity to take advantage of different energy sources. 

Polyphagia is an evolutionary rarity that is unequally distributed in herbivorous groups [3]. Only 2% of herbivorous insects are polyphagous, suggesting that evolution favors dietary specialism [4]. Transcriptome studies have provided insights into how insect herbivores attain polyphagy, implicating transcriptional regulation of the genes coding proteins involved in the digestive system [3,5]. Generalist insects can regulate genes belonging to three main categories according to dietary needs: ribosomal, detoxifying, and digestive genes [6]. 

The set of genes encoding digestive enzymes has been suggested as an important determinant of the successful conquest of a wide range of hosts. We have shown previously that generalist species have a greater repertoire of genes encoding digestive enzymes and greater transcriptional plasticity than specialist insects [7,8,9]. Digestive enzymes such as proteases are dynamically regulated in response to the food source [10], which seems to be a common feature of the family Noctuidae [5]. However, because insect proteases are so important for host plant utilization, many plants have evolved protease/peptidase inhibitors (PIs) as an antiherbivore defense mechanism. Insects have in turn developed mechanisms to counteract the antinutritional effects of these defensive PIs. These mechanisms include the upregulation of existing digestive proteases or inhibitor-insensitive protease isoforms and/or the activation of proteases that metabolize plant inhibitors [11]. Although significant progress has been made in understanding the molecular mechanisms of insect adaptation to PIs, the suite of genes involved and how these genes are regulated remains a topic of ongoing research.

In the current study, we addressed this knowledge gap by investigating the molecular mechanisms involved in the resistance of *H. armigera* larvae to soybean peptidase inhibitors (SPIs). To this end, we analyzed the transcriptional and DNA methylation profile of larvae fed on SPIs and their progeny reared on an inhibitor-free diet and contrasted these data with those of larvae reared on a control diet. Our analyses revealed large-scale changes in gene expression associated with feeding on a diet supplemented with SPIs, including the upregulation of genes encoding serine peptidases. A major component of the transcriptional signature induced by SPIs was also identified in the larvae of the next generation, which were never exposed to SPIs.

## 2. Results

### 2.1. H. armigera Larvae Are Resistant to SPIs

The semipurified extract of SPI showed a predominant inhibitory action against trypsins (Appendix A). The concentration of inhibitors was estimated at 11 and 1.4 mg of trypsin and chymotrypsin inhibitors, respectively. The diet supplemented with SPI did not cause any substantial negative effect on the analyzed biological parameters of the tested *H. armigera* (Figure 1A). More specifically, feeding on a diet containing SPI did not significantly affect the pupae weight, insect survival rate, the number of females laying eggs, or the number of eggs per female (Figure 1B–D).

### 2.2. SPI Ingestion Results in Marked Changes in Gene Expression in the H. armigera Midgut

A total of 839 genes were differentially expressed in the *H. armigera* larvae midguts exposed to SPIs (Appendix A). A GO enrichment analysis revealed 11 and 3 enriched GO terms associated with downregulated and upregulated genes, respectively (Figure 2A,B). Enriched GO terms in downregulated genes in the progeny of SPI-treated larvae were associated with cell motility, the mitochondrial inner membrane complex, ATP transport and biosynthesis processes, the phosphorus metabolic process, and glutamine family amino acid biosynthesis (Figure 2A). Enriched GO terms in upregulated genes represented serine-type endopeptidase activity (Figure 2B). For downregulated gene expression that persisted in the progeny, the enriched GO terms were associated with cell motility, phosphatase binding proteins, and pyruvate kinase activity (Figure 2C). 

### 2.3. Differential Expression of Digestive Peptidases May Be Associated with the Adaptation of H. armigera to SPIs

Feeding larvae on a diet containing SPIs altered the transcription level of 32 genes encoding primarily proteolytic enzymes (Figure 3A). A total of 31 trypsin and 23 chymotrypsin genes were identified as having a complete catalytic triad, of which eight were activated to overcome the antinutritional effect of SPIs (Appendix A). Serine peptidase homologous genes, probably without encoded catalytic activity, were upregulated (LOC110379025; XP_049693481.1 without His57 and Ser195) and downregulated (LOC110384681; XP_021201764.1 without Ser195) in *H. armigera* guts by SPI feeding (Appendix A). Similarly, three transmembrane protease serine genes were downregulated, while the level of transcripts of one transmembrane protease serine increased (Appendix A). A Phylogenetic analysis showed that active serine peptidase genes tended to be more closely related and separate from downregulated genes, suggesting some homology in the enzyme primary structures. In parallel, at least 36 genes potentially involved in regulatory processes were repressed in the larvae exposed to SPIs and their progeny (Figure 3B). Downregulated genes in the presence of SPIs may be associated with gene regulatory mechanisms, controlling transcript synthesis in the digestive system of *H. armigera*. Candidates for regulatory processes include genes that can act as elongation factors (ex. 1-alpha 2) or transcription factors (ex. NF-kappa-B) (Figure 3B).

### 2.4. Transgenerational Persistence of the Transcriptional Profile Induced by SPI Exposure

The gene expression profile of the second generation of *H. armigera* from larvae exposed to SPIs was found to be more similar to that of first-generation larvae fed with SPIs (Figure 4 and Figure 5) than first-generation larvae fed on a diet lacking SPIs. This indicated that the expression pattern of many of the differentially expressed genes in the first SPI generation persisted in the progeny (Figure 4 and Figure 5). Differentially expressed transcripts of the second generation (from larvae exposed to SPI) matched with the larvae of the first generation exposed to SPIs in 59.2% (potential epialleles) and 0% of cases when compared with larvae of the first generation fed on an artificial diet without SPIs (Figure 4). As expected, five upregulated genes encoded trypsins and two were precursors of oxidoreductase enzymes (Table 1). Epialleles of gene expression patterns were selected for validation by RT-qPCR. This revealed that the results of the RNA-seq gene expression analysis were consistent with the RT-qPCR analysis for the three trypsin-encoding genes and three regulatory function genes tested (Figure 5).

### 2.5. DNA Methylation May Be Involved in the Molecular Response of H. armigera to SPI Ingestion

The characterization of the midgut cell methylome identified a total of 99 genes with differentially methylated loci (Appendix A) that may be involved in the molecular response to SPIs. Of these, 19 genes preserved the epigenetic mark in the second generation even though the larvae were never directly exposed to SPIs (Table 2). Three hypomethylated genes in the presence of SPIs and in the second generation appeared to be involved in metabolism of B vitamins, including thiamine transporter 2, biotin-protein ligase, and bifunctional methylenetetrahydrofolate dehydrogenase/cyclohydrolase 2 (Appendix A). Similarly, genes that could be involved in transcriptional regulation and protein post-translational modification were hypomethylated in both treatments compared to larvae fed on an SPI-free diet, including ubiquitin carboxyl-terminal hydrolase 3, lipoma-preferred partner, zinc finger SWIM domain-containing protein 8, and transcriptional repressor CTCF-like. In contrast, hypermethylated genes exclusively identified in SPI-fed larvae and in the second generation could be associated with transport functions and translation elongation factors such as tumor suppressor candidate 3, elongation factor 1-alpha, coatomer subunit alpha, and mucolipin-3 (Appendix A).

We had a special interest in studying the DNA methylation levels in trypsin and chymotrypsin genes, most of which are associated with CG islands (Appendix A). Genes encoding serine peptidases were almost completely unmethylated in all treatments (Appendix A). CG islands were resistant to trypsin DNA methylation even in adult tissues (Appendix A). BSP and WGBS results revealed no direct relationship between DNA methylation and the transcriptional regulation of serine peptidase genes.

## 3. Discussion

Our data provided evidence of the role of key proteins in SPI adaptation, revealing 32 proteolytic enzymes differentially expressed in larvae exposed to SPIs. The transcriptional plasticity of proteolytic enzyme genes is a conserved mechanism in lepidopterans such as *H. armigera* [5,9,10,12,13] and may be involved in the successful utilization of multiple host plants. The dynamic regulation of digestive enzymes may be the most important factor in adaptation to SPI, since this has been shown to be a common feature in generalist insects [5,7,8,9]. Recent work provided evidence that *H. armigera* has functionally redundant paralogs of serine peptidase genes that allow it to maintain the metabolic robustness of the digestive system [14]. Our transcriptome profiling revealed that genes encoding 17 proteolytic enzymes were reduced following SPI exposure. These enzymes may include serine peptidases that are sensitive to SPIs or those that are influenced by changes in the content of free amino acids in the digestive system, as suggested for *Locusta migratoria* by Spit et al. [15]. In contrast, the 15 genes upregulated in *H. armigera* following SPI exposure may include serine peptidases that are resistant to SPIs (Figure 6).

The functional characterization of catalytic proteins will contribute to understanding how polyphagous insects overcome plant defense molecules. *S. frugiperda* larvae exposed to SPIs activate a set of related genes that encode enzymes with a specific amino acid composition; these are absent in specialist insects such as *D. saccharalis* [5]. These genes encode trypsin and chymotrypsin enzymes with conserved residues at positions associated with substrate binding such as Arg188, Gln190, and Phe215 [5]. These residues were not conserved in the *H. armigera* homologs even though some orthologs grouped in the same clade (SfChy8, SfChy9, SfChy21, HaChy XP_021201464.1, and Ha Transmembrane protease serine 9 XP_049700702.1), suggestive of a degree of functional conservation between them (Appendix A). *S. frugiperda* larvae responded to SPI exposure by activating a larger group of chymotrypsins with at least seven upregulated genes, while in the current study, only two chymotrypsin genes and one transmembrane protease serine gene were upregulated in SPI-exposed *H. argmigera*. This suggested that the underlying molecular mechanisms of SPI adaptation could vary in different members of the family Noctuidae.

The SPI treatment stimulated the expression of the one transmembrane protease serine gene and reduced the expression of three other homologues that preserve the catalytic structure of trypsins (Appendix A). The activated gene (LOC110384488; XP_049700702.1) and one of the repressed genes (LOC110380360; XP_049693306) encode an unusual peptidase with two complete trypsin domains (IPR001254). Trypsin multidomain enzymes, also known as polytrypsins, are found in humans, rodents [17], and *Drosophila* [18]; it has been suggested that these enzymes have a greater proteolytic capacity compared to single-domain trypsins [19]. According to our RNA-seq results (Appendix A), the relative expression values of the polytrypsin gene were among the highest (2.1 to 4.7 Log2-fold change) of the genes overexpressed following SPI exposure, and the repressed polytrypsin gene was among the most strongly downregulated, along with the transmembrane protein and carboxypeptidase B genes (−2.8 to −3.5 Log2-fold change).

Hydrolytic enzymes and trypsins play a key role in the primary digestion of food and other substances. *H. armigera* larvae exposed to phytotoxin stress secreted enzymes that included hydrolases, serine proteases, and lipases [20]. Similarly, SPIs activated genes associated with fatty acid catalysis and others involved in the redox system, including those encoding lipases, glutathione S-transferases, peroxidases, uricase and aldehyde oxidase (Appendix A). Thus, the regulation of genes involved in detoxification and digestion may be intimately linked [20,21]. Primary metabolism genes such as 4-coumarate–CoA ligase-like 9 and sorbitol dehydrogenase-like were upregulated in *H. armigera* larvae exposed to SPIs (Appendix A). This was consistent with previous work demonstrating that SPI ingestion impacts the regulation of genes involved in carbohydrate and lipid metabolisms in *H. armigera*, in addition to genes encoding digestive and detoxification enzymes [10,22]. A similar pattern was observed in the generalist herbivore *L. migratoria* [15]. This response may be related to the requirement to modulate energetic metabolism to compensate for the metabolic cost incurred from peptide degradation.

Intriguingly, our data revealed that genes encoding serine proteases with incomplete catalytic triad were also regulated by SPIs (Appendix A). The role of catalytically inactive serine protease homologs in *H. armigera* digestion is unclear. However, knockout of a cluster of 18 trypsin-like genes in *H. armigera* demonstrated that serine peptidase genes and homologous genes showed coordinated patterns of expression, suggestive of a coregulatory gene network that maintains metabolic and catalytic equilibrium [14]. In lepidopterans, serine protease homologs participate as modulators of proteolytic enzymes [23,24], immune responses [25,26], and development [27,28]. In future studies, the causal role of the proteolytic enzymes and homologous proteins identified in this study in mediating SPI tolerance should be examined; for example, by functionally expressing them in vitro and/or by gene knockout using CRISPR-Cas9 genome editing.

The molecular mechanisms that control the regulation of serine peptidases remain unresolved [12,16,29]. Our gene expression profiling uncovered candidates that may be involved in the regulation of serine peptidases by identifying 36 differentially expressed genes that have regulatory functions (Figure 3B). The molecular response to SPI may be controlled at both the pretranscriptional and post-transcriptional levels, as illustrated by the molecular mechanisms of *Galleria mellonella* resistance to *Bacillus thuringiensis* [30]. In this regard, we also highlighted genes that play a role in epigenetic mechanisms such as potential histone modifiers and proteins with DNA-binding domains. For instance, proteins with zinc domains can recruit chromatin remodelers, as has been demonstrated in *Drosophila* [31] and *Bombyx mori* [32]. *H. armigera* has at least 398 loci encoding proteins with zinc domains, but the functions of almost all are completely unknown in most insects [33]. 

A large part of the transcriptional profile observed in *H. armigera* larvae fed with SPIs persisted in its progeny that had not been directly exposed to SPIs. Epigenetic mechanisms may be involved in the expression of genes in the digestive system and the formation of epialleles in *H. armigera*. Previous studies showed that the expressed protease profile depends on the type of diet and is partially transgenerationally maintained in *S. frugiperda* [34] and *S. exigua* [35]. Related to this, there is growing evidence of the role of epialleles in insect adaptation. A wide range of stimuli can induce a phenotypic response that can be inherited, even for more than one generation. For instance, the F1 and F2 offspring of *Aedes albopictus* mosquitoes exposed to the phytoestrogen genistein and the fungicide vinclozolin show a reduced sensitivity to the insecticide imidacloprid [36] with DNA methylation patterns formed in the genome of parental insects stably inherited in the progeny. 

Diet is one of the main factors that causes epigenetic changes in insects [37,38] and other animals, affecting DNA methylation and histone acetylation patterns [39,40]. SPIs have been proposed as a potential control of insect pests by incorporating them into their diets [13,41,42,43,44,45,46] or increasing their concentration in plants [47,48,49,50]. However, our data suggested that the upregulation of digestive-related proteins may allow the inhibitory properties of SPIs in *H. armigera* to be counteracted. Furthermore, these gene expression patterns may persist intergenerationally. Here, it was not possible to identify significantly differentially methylated genes with proteolytic functions; the transcriptional plasticity of the digestive system may depend upon cross-talk between regulatory mechanisms. Recent research indicated that DNA methylation in insects was functionally linked to other epigenetic marks [51]. In our results, at least 32 differentially methylated genes with regulatory functions were found in the genome of larvae exposed to SPIs, many of which raised intriguing questions (Appendix A). However, the regulation of transcript synthesis of the digestive system may depend on several regulatory factors, including transcriptional and translational control processes [52,53,54]. 

In conclusion, we provided evidence that the differential regulation of digestive enzymes plays a role in the adaptation of larvae to the antinutritional effects of SPIs. Larvae exposed to SPIs activated a set of proteolytic genes (predominantly trypsins). RNA-seq and Methyl-seq analyses of the digestive system of the progeny of the insects exposed to SPIs suggested that a large proportion of gene expression changes may have been transmitted to the next generation even in the absence of antinutritional molecules in the diet of progeny. However, we did not identify a role for gene body methylation changes in the genes identified, and it remains to be seen if this phenomenon is mediated by methylation changes of other genes in a regulatory pathway or by other epigenetic mechanisms such as histone modifications or ncRNAs. At least in polyphagous insects, inheritance of acquired molecular phenotypes may occur before the emergence of new genetic polymorphisms that support some adaptive advantage. Here, we showed a set of genes that may be involved in the molecular response to SPIs and that will be important targets in future studies.

## 4. Materials and Methods

### 4.1. Extraction of Soybean Proteinase Inhibitors (SPIs)

SPIs were extracted from soybean (*Glycine max* L.) seeds of the *Potência* variety following the methodology described by Souza et al. [5] and Paulillo et al. [7]. A total of 100 g of seeds was crushed in 1 L of saline solution (0.15 M NaCl) for 30 min at room temperature. The solution was centrifuged at 3000× *g* for 20 min at 4 °C. The supernatant was collected, and the protein extract was precipitated with cold acetone (70% *v*/*v*) by centrifugation at 6000× *g* for 20 min at 4 °C. The extracts enriched with SPIs were lyophilized for 72 h and stored until needed for bioassays.

### 4.2. Inhibitory Activity of SPIs

The inhibitory activity analysis was carried out according to Hummel [55]. The N-4-tosyl-L-arginine methyl ester (TAME) and N-benzoyl-L-tyrosine ethyl ester (BTEE) compounds were used as specific substrates for trypsin and chymotrypsin, respectively. For the trypsin-like activity assay, the standard assay mixture contained 100 µL of trypsin (15.4 µg/mL 1 mM HCl), 2.6 mL of Tris hydrochloride buffer (46 mM, pH 8.1 with 11.5 mM CaCl2), 300 µL of TAME (10 mM), and 0–100 µL of the semipurified extract of SPIs (1 mg/mL). For chymotrypsin-like activity, the standard assay mixture contained 100 µL of chymotrypsin (15.4 µg/mL 1 mM HCl), 1.5 mL of Tris hydrochloride buffer (80 mM, pH 7.4 with 100 mM CaCl_2_), 1.4 mL of BTEE (1.07 mM), and 0–100 µL of the semipurified extract of SPIs. Measurements were made using quartz cuvettes in an accuSkan GO spectrophotometer (Thermo Scientific, Waltham, MA, USA). Absorbance differences at 247 and 256 nm were measured at intervals of 30 s over 10 min to estimate the rate of hydrolysis of the TAME and BTEE, respectively. The total proteolytic activity was determined by performing the same assay using samples without SPIs.

### 4.3. Insect Rearing and Feeding Assays

The *H. armigera* colony used in this study was established with specimens from the Insect Biology Laboratory of Luiz de Queiroz College of Agriculture, University of São Paulo (ESALQ/USP). The insects were maintained at 25 ± 1 °C in 60–70% relative humidity and a 14:10 h (L:D) photoperiod. 

A total of 300 neonate larvae were transferred into glass tubes containing an artificial diet [56] supplemented with or without 0.5% (*w*/*v*) SPIs (Appendix A). The larvae were checked daily for survival until the last instar. The pupae were removed from the glass tubes and placed in Petri dishes (15 × 2 cm) with a filter paper disc that was moistened daily. The pupal weight and sex ratio were determined 24 h after pupation. A total of 25 couples per treatment were separated in 100 mm PCV tubes with the upper end of the cage covered with tulle. Adults were fed with a 10% honey solution. The egg number per female and the longevity and survival of the pupae and moths were recorded daily. The statistical significance (*p* < 0.05) of the results were assessed using analysis of variance (ANOVA) and Tukey’s post hoc test. 

Another group of larvae were kept on the same type of diet until the adult stage in order to obtain a second generation. The tissue with the eggs was placed in clear plastic cups (500 mL) together with a piece of filter paper moistened with distilled water. The cups were kept in climate-controlled chambers at 25 °C and in a photoperiod of 14:10 h (day:night) until the larvae hatched. Newly hatched larvae were again transferred to artificial diet treatments with and without SPIs until reaching the last instar. Guts were collected from insect progeny that were subjected to SPIs for the transcriptome and methylome analysis.

### 4.4. RNA-Seq Analysis

Transcriptional changes in the digestive system of larvae exposed to SPIs were examined using transcriptome sequencing. Neonates were fed both types of diet (Appendix A) until the 6th instar. Midguts were extracted from 30 randomly selected larvae from each treatment and stored at −80 °C. Another group of larvae were reared to adults as described above. Eggs from insects exposed to SPIs were randomly selected and placed in Petri dishes with moistened filter paper. Newly hatched larvae were fed an artificial diet without SPIs until the 6th instar. Once again, midguts were extracted from 30 random larvae and stored at −80 °C.

Total RNA was isolated from pooled samples of 10 midguts per biological replicate with 3 replicates per treatment. The samples were ground in liquid nitrogen and the RNA was extracted using an RNeasy Mini Kit (Qiagen); DNA was removed using the RNase-Free DNase Set (Qiagen) following the manufacturer’s protocol. The RNA quality and yield were confirmed using an Agilent 2100 bioanalyzer instrument (Agilent Technologies, CA, USA). The RNA-seq libraries were prepared using the TruSeq Stranded mRNA Sample Preparation kit (Illumina, Inc.).

Sequencing was performed on the Illumina HiSeq 2500 v4 platform using the paired-end protocol and a 100-base-pair read metric. Quality control analysis was performed using the FastQC V 0.11.9 program [57]. Preprocessing of the data such as trimming of adapters was performed with the Trimmomatic V 0.32 program [58]. The mapping of reads to the reference genome of *H. armigera* deposited in the NCBI database (Accession ID: GCA_002156985.1) was performed with the STAR v2.7.0 program [59] using the default parameters. In order to increase the consistency and integrity of the RNA-seq data, we discarded two samples that showed a significant deviation from other replicates (Appendix A). Differentially expressed genes were identified using the DESeq2 package [60] in R v4.0.5 (https://www.r-project.org). Genes were considered significantly differentially expressed when *p-adjust* < 0.05 and log2 fold-change >1 or <−1. The differentially expressed genes identified were used to construct Venn diagrams in R software v4.0.5 (https://www.r-project.org). Graphical abstracts and Figure 6 were generated with BioRender (https://biorender.com/).

### 4.5. Identification of Serine Peptidase Gene Sequences

Serine peptidase genes present in the *H. armigera* genome were identified using the BLASTP program with a set of well-described SP genes as the query [5]. The identified sequences were then filtered using the presence of a predicted trypsin domain (IPR001254) according to an InterproScan analysis and a complete catalytic triad (His57, Asp102, and Ser195) [61]. Subsequently, trypsin and chymotrypsin genes were classified according to the amino acid at position 189 (Asp for trypsin). A multiple sequence alignment was performed in MAFFT v7.0 [62], and a phylogenetic tree was generated using IQ-TREE v1.6.12 [63]. The phylogenetic tree was visualized using the Interactive Tree of Life (iTOL) tool v6.5.8 [64]. For comparison, the amino acid sequences of trypsins and chymotrypsins of *S. frugiperda* reported by Souza et al. [5] were included in the phylogenetic analysis.

### 4.6. CpG Island Mapping 

The mapping of CpGs was carried out on trypsin and chymotrypsin gene sequences as well as regulatory regions. The sequences were analyzed using the CpG prediction program CpGPLOT [65]. The analysis was carried out using the default parameters: with an average of 10 windows in not less than 200 bp or calculated content (%G + %C) greater than 50% and a calculated observed/expected ratio greater than 0.6 [66]. In the same way, they were mapped to CpG islands in regulatory regions (5 kb upstream and downstream of genes of interest). Finally, a descriptive analysis of the position, size, and frequency of CpG islands was conducted.

### 4.7. Functional Analysis of DNA Methylation in CpG Islands of Serine Peptidase Genes

The study of DNA methylation in CpG islands was evaluated using DNA sequencing treated with sodium bisulfite and PCR (BSP, bisulfite sequencing PCR), using a protocol similar to the methodology described by Xu et al. [51]. Genomic DNA was isolated from midgut samples of first-generation larvae (fed on a diet supplemented with or without SPIs) and from progenies of larvae exposed to SPIs. Each treatment had three experimental replicates for a total of nine experimental units. Groups of 10 midgut samples from each treatment were macerated in liquid nitrogen. Genomic DNA was isolated using the DNeasy Blood and Tissue Kit (QIAGEN).

To study the role of DNA methylation in the CpG islands of serine peptidase genes in insect development, we used tissue from the legs, head, and thorax of adults of *H. armigera*. Methylation profiles were compared with DNA methylation patterns from *H. armigera* larvae fed with and without SPIs. Unmethylated cytosines from 1 μg of DNA were converted to uracil using the EpiJET Bisulfite Conversion Kit (Thermo Scientific) following the manufacturer’s instructions. The primers were designed to amplify CpG islands using the MethPrimer program following the standard parameters of the Pick primers for bisulfite sequencing PCR software (Appendix A) [67]. Three SPI-responsive serine peptidase genes were selected for BSP analysis using degenerate primers. Amplification reactions were performed using 5 ng of bisulfite-treated DNA for 40 cycles (95 °C for 30 s and 72 °C for 2 min). The amplified fragments were analyzed on a 2% agarose gel. Finally, PCR products were sequenced to identify methylated cytosines. Differential methylation analysis of CG dinucleotides was performed using the BISMA program [68].

### 4.8. Genome-Wide Characterization of DNA Methylation

Genomic DNA was isolated from midgut samples from first-generation *H. armigera* larvae (fed either an SPI-supplemented or control diet) and from progenies of larvae fed the SPI-supplemented diet using three replicates per treatment. Samples were shipped to Novogene (UK) Company Ltd. for library preparation, bisulfite conversion, and sequencing on the Illumina NovaSeq platform (150 bp paired end reads). Quality control was performed using Trimmomatic v0.38, including a hard trimming of the 5 region of the paired reads (—clip_R1 10—clip_R2 25) due to library-preparation artefacts. Reads were aligned to the reference genome of *H. armigera* (Accession ID: GCA_002156985.1) using Bismark v0.22.1 in paired mode with the options “—score_min L,0,−0.4 –unmapped”. Reads that did not map in paired mode were subsequently mapped in single-end mode using “—score_min L,0,−0.4”. Alignment files were deduplicated and the methylation calls were extracted and destranded using Bismark scripts. The CpG output files were imported into MethylKit v1.8.1 in R version 3.5.2; we filtered sites with coverage < 10 or above the 99.9th percentile. Each site was tested against a binomial distribution to identify sites with significant levels of methylation; those unmethylated in all samples were filtered out. The remaining loci were pooled per treatment to combine coverage and subjected to pairwise tests between groups using Fisher’s test and corrected for multiple testing using the SLIM method. Loci with a methylation difference of at least 25% and *q-adjusted* by FDR 0.01 were considered differentially methylated.

### 4.9. RT-qPCR Experiments

Validation of expression profiling was carried out using transcription-quantitative PCR (RT-qPCR). The cDNA was synthesized using reverse transcription with 1 µg of RNA, oligo d(T) primers, and reverse transcriptase enzyme (Promega) following the manufacturer’s instructions. RT-qPCR reactions were performed in 25 µL containing 0.3 μL of forward primer (0.1 μM) (Appendix A), 0.3 μL of reverse primer (0.1 μM) (Appendix A), 3 μL of cDNA, and 12.5 μL of SYBR^®^ Green/ROX qPCR Master Mix (2X) (Fermentas). The reactions were carried out in a thermocycler StepOne™ Real-Time PCR System (Applied Biosystems) with reaction conditions comprising 50 °C for 2 min and 95 °C for 10 min followed by 40 cycles of 95 °C for 10 s and 60 °C for 30 s. After 40 cycles, a melt-curve step was performed by heating the reaction mixture from 60 to 95 °C at a rate of 1 °C/s. A negative control without cDNA was included in each analysis. L27 and S18 ribosomal protein genes were used to normalize gene expression data [10,69]. The amplification efficiency was determined using the LinReg PCR software [70], and relative quantification was carried out using the REST^®^ software [71].

### 4.10. Functional Annotation and Gene Ontology Enrichment

The *H. armigera* protein dataset was annotated with BLASTP against the Arthropoda subset of the NCBI NR database and then subjected to GO term mapping and an InterProScan analysis using BLAST2GO and OmicsBox [72]. GO enrichment analyses of significantly differentially expressed or methylated sites using Fisher’s test were conducted with the same software. All analyses used the default settings; enriched GO results were reduced to the most specific prior to visualization.

## Figures and Tables

**Figure 1 ijms-23-14301-f001:**
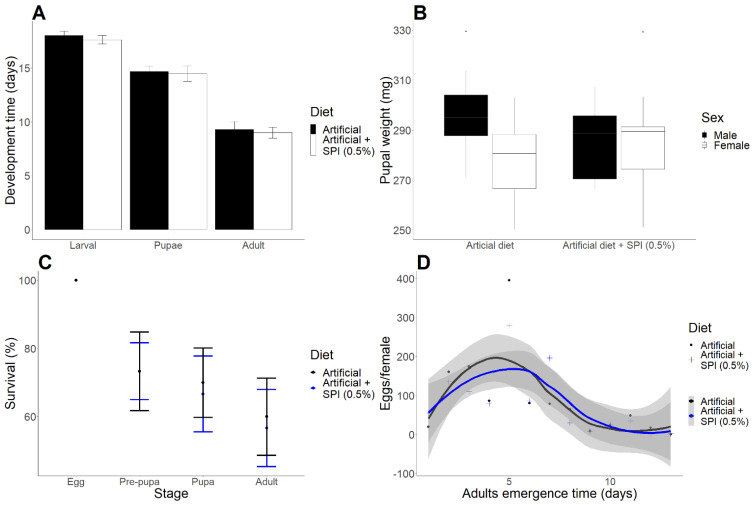
Influence of diet supplemented with SPIs (0.5%) on development time (**A**), pupal weight (**B**), survival (**C**) and number of eggs per female (**D**) of *H. armigera*. Error bars represent standard error (**A**–**C**). The shadings in (**D**) indicate 80% confidence intervals.

**Figure 2 ijms-23-14301-f002:**
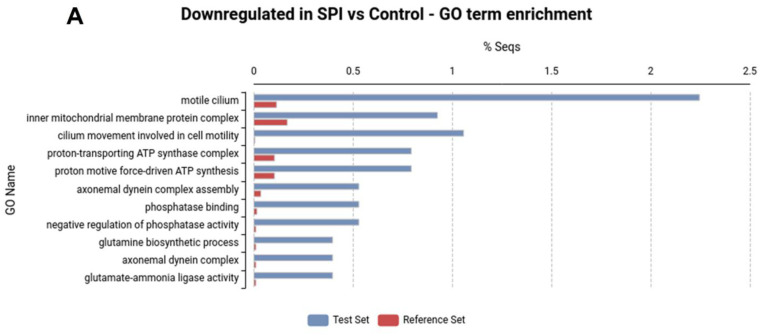
Enrichment analysis showing percentage of sequences associated with over-represented GO terms out of the genes that were (**A**) downregulated, (**B**) upregulated in the digestive system of *H. armigera* by the presence of SPIs in the artificial diet, and (**C**) downregulated in the digestive systems of *H. armigera* progeny that were exposed to SPIs in comparison to reference set (all genes expressed).

**Figure 3 ijms-23-14301-f003:**
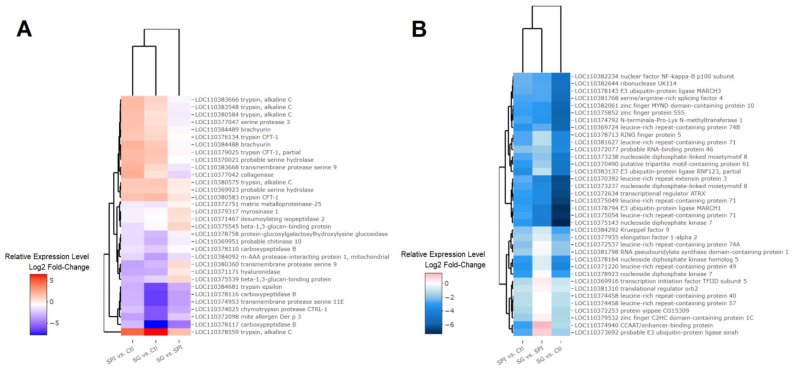
Heatmap and clustering of differentially expressed genes (Log2-FoldChange values >1 or <−1 and *p*-adjusted <0.05) encoding digestive enzymes (**A**) and regulatory functions (**B**) in the gut of *H. armigera* after exposure to SPIs (SPI vs. Ctl). The transcriptional profile of the second generation were compared with larvae from the first generation fed with and without SPIs (SG vs. SPI and SG vs. Ctl, respectively).

**Figure 4 ijms-23-14301-f004:**
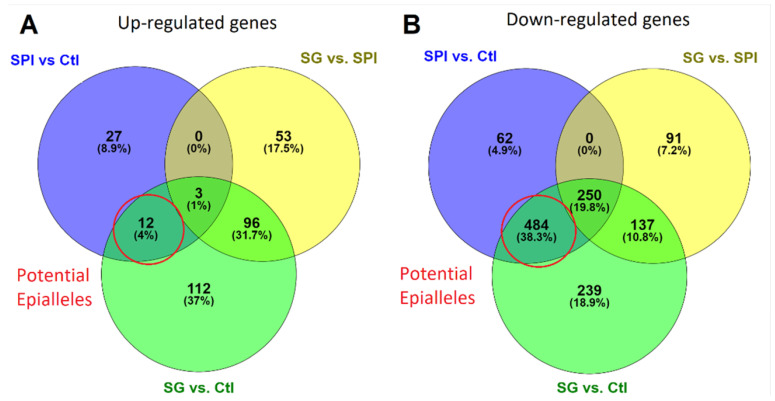
Venn diagram of the differentially expressed genes between three comparisons for upregulated (**A**) and downregulated genes (**B**). The number of differentially expressed genes of the first generation of larvae fed with and without SPIs (SPI vs. Ctl). The number of differentially expressed genes of the second generation (F1) compared to first generation of larvae fed on artificial diet with SPIs (SG vs. SPI) and without SPIs (SG vs. Ctl). We considered as potential epialleles the genes that were induced by SPIs and had the same response in the second generation of larvae fed a control diet without SPIs.

**Figure 5 ijms-23-14301-f005:**
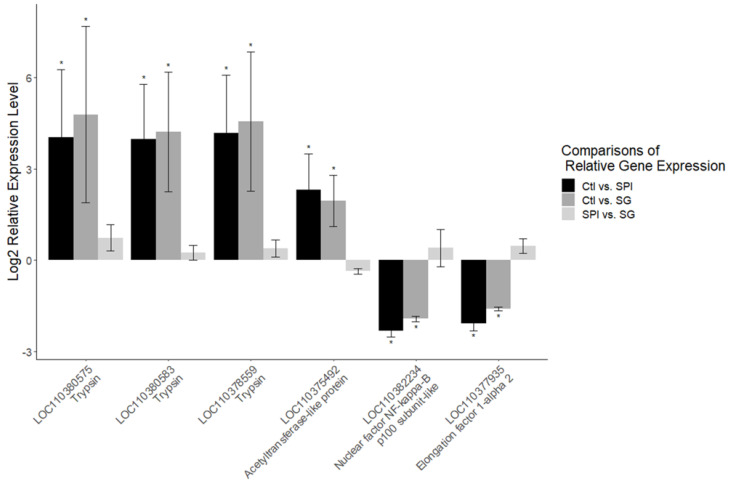
Validation by RT-qPCR of potential epialleles of gene expression patterns induced in *H. armigera* larvae by SPIs. * Significant differences (Log2 > 1 or < −1, *p* < 0.05) for differentially expressed genes.

**Figure 6 ijms-23-14301-f006:**
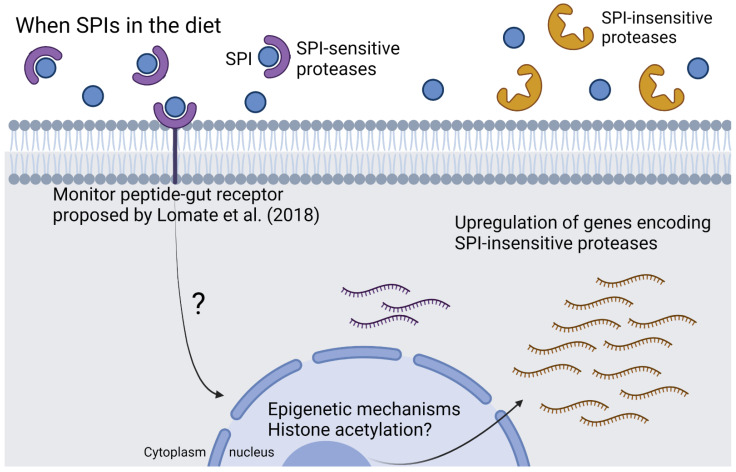
Proposed mechanism of protease regulation by epigenetic mechanisms in *H. armigera* larvae fed with SPI. According to Lomate et al. [16], the monitor peptides are present in insect gut cells and could regulate digestive protease expression. The information on the presence of SPIs in the diet reaches the nucleus through a molecular mechanism that is still unknown. Epigenetic mechanisms could be controlling the dynamic regulation of genes encoding serine proteases to adapt to the antinutritional effect of SPIs.

**Table 1 ijms-23-14301-t001:** Potential epialleles of upregulated genes induced by SPIs in *H. armigera* larvae.

Locus	SPI vs. Ctl		SG vs. Ctl		Protein Name
	Log2 Fold-Change	(*p-Adjust* < 0.05)	Log2 Fold-Change	(*p-Adjust* < 0.05)	
LOC110370568	1.03	0.002	1.76	0.000	Diuretic hormone
LOC110371086	1.03	0.000	2.48	0.000	Uricase
LOC110372133	2.03	0.000	1.50	0.012	Probable aldehyde oxidase 2
LOC110375492	1.16	0.012	1.81	0.000	Uncharacterized protein LOC110375492
LOC110377152	1.13	0.020	1.59	0.001	Uncharacterized protein LOC110377152
LOC110378559	4.75	0.036	6.43	0.003	Trypsin, alkaline C-like
LOC110378803	1.37	0.035	1.47	0.025	Gloverin-like
LOC110379025	1.90	0.000	1.48	0.009	Trypsin CFT-1-like, partial
LOC110380575	1.33	0.001	1.40	0.001	Trypsin, alkaline C-like
LOC110380583	1.59	0.000	1.90	0.000	Trypsin CFT-1-like
LOC110383800	1.87	0.000	1.69	0.001	Lipoprotein lipase-like
LOC110384494	1.13	0.022	1.08	0.040	Serine proteases, trypsin domain (IPR001254)

**Table 2 ijms-23-14301-t002:** Potential DNA methylation epialleles induced by SPIs in the genome of *H. armigera*.

Locus	Ctl vs. SPI	qvalue	Protein	Ctl vs. SG	qvalue
LOC110384175	−28.2	0.01	Uncharacterized LOC110384175	−29.0	0.00
LOC110369926	−30.5	0.00	Pre-rRNA processing protein FTSJ3	−40.7	0.00
LOC110369936	−29.8	0.01	A-kinase anchor protein 10, mitochondrial	−29.6	0.00
LOC110370916	−34.3	0.00	Cytochrome b-c1 complex subunit 6, mitochondrial-like	−26.5	0.01
LOC110371715	−32.4	0.00	Microprocessor complex subunit DGCR8, transcript variant X1	−28.6	0.00
LOC110372825	−39.1	0.00	THO complex subunit 4	−35.0	0.00
LOC110372886	−31.5	0.01	Beta-arrestin−1, transcript variant X1	−28.3	0.00
LOC110374361	−33.5	0.00	Regulator of nonsense transcripts 1 homolog	−31.4	0.00
LOC110374662	−36.2	0.00	nudC domain-containing protein 1	−30.2	0.00
LOC110374778	−35.9	0.01	Acetyl-coenzyme A transporter 1	32.9	0.00
LOC110375095	34.5	0.00	Uncharacterized LOC110375095	31.5	0.00
LOC110375190	38.0	0.00	Uncharacterized LOC110375190	28.6	0.01
LOC110375794	37.3	0.00	Voltage-dependent anion-selective channel-like	32.3	0.00
LOC110377217	−25.2	0.01	Uncharacterized LOC110377217	−32.9	0.00
LOC110380153	−25.6	0.00	Uncharacterized LOC110380153	−25.6	0.00
LOC110381188	−29.3	0.00	Uncharacterized LOC110381188	25.4	0.00
LOC110382008	30.0	0.01	Zinc finger protein 2-like	30.7	0.00
LOC110383013	34.6	0.01	Thioredoxin-related transmembrane protein 2 homolog, transcript variant X1	27.9	0.00
LOC110383900	−27.7	0.00	Thiamine transporter 2-like	−30.8	0.00

## Data Availability

The RNA and bisulfite sequencing data generated in this study have been deposited in the NCBI SRA under the BioProject accession number PRJNA884530.

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
