# Peer review of "Adaptation of *Helicoverpa armigera* to Soybean Peptidase Inhibitors Is Associated with the Transgenerational Upregulation of Serine Peptidases"

_ijms, 2022, doi:10.3390/ijms232214301_

Round 1

Reviewer 1 Report

The manuscript is very well written and presented.

However, once the scope of the study was to identify the mechanism, I would like to suggest to design an mechanism. It will give a greater value to the study

Author Response

Dear Editor,

We have addressed all the points raised by the referee. Thanks for this improvement.

Reviewer 2 Report

The manuscript entitled "Adaptation of Helicoverpa armigera to soybean peptidase inhibitors is associated with the transgenerational upregulation 3 of serine peptidases" by Velasquez-Vasconez et al studied transcriptional and methylation changes of H. armigera fed by soybean peptidase inhibitors (SPI). Potential epialleles were identified that transferable to second generation.  Comparative analysis didn't show a direct relationship between between differentially methylated and differentially expressed genes.  The authors further found DNA methylation potential epialleles were associated with transcriptional and translational controls, and these may play a regulatory role in H. armigera adaptation to SPIs. I have some concerns about experimental design before its acceptance.

Comments

1. The Extraction of Soybean Proteinase Inhibitors (SPI) procedure is the extraction of total protein in soybean seeds to me. Please specify how SPI were enriched during extraction, and provide citations. Besides, the acetone used in the isolation procedure may left in the extracts that can be a confounding factor in this experiment. 

2. in the Material and Method, Inhibitory activity of SPIs  section, 'For chymotrypsin-like activity, the standard assay mixture contained 100 μl trypsin (15.4 μg/ml 334 1 mM HCl), 1.5 ml Tris hydrochloride buffer (80 mM, pH 7.4 with 100 mM CaCl2), 1.4 ml 335 BTEE (1.07 mM) and 0-100 μl of the semi-purified extract of SPI. '  (Line 333-335), why use trypsin instead of chymotrypsin for chymotrypsin activity analysis? 

3. I didn't see any supplementary Figures attached, please provide Figure S1-S6.

4. How 0.5% SPI was determined to use as diet supplement in this research?

5. Figure 2, please clarify '%seq', 'Test set' and 'Reference set'.

6. Figure 3. I assume 'IPS' is 'SPI'? 

Author Response

Dear Editor,

We have now a newly-improved version of the manuscript, addressing all the issues pointed out by the referee. I would like to thank all the reviewer's comments.

Round 2

Reviewer 2 Report

The authors basically addressed my concerns in the revised manuscript. I don't have further concerns for its publication.